# Comparative Insights into Acute Gastroenteritis in Cattle Caused by Bovine Rotavirus A and Bovine Coronavirus

**DOI:** 10.3390/vetsci11120671

**Published:** 2024-12-21

**Authors:** Vjekoslava Kostanić, Valentina Kunić, Marina Prišlin Šimac, Marica Lolić, Tomislav Sukalić, Dragan Brnić

**Affiliations:** 1Department of Virology, Croatian Veterinary Institute, 10000 Zagreb, Croatia; kostanic@veinst.hr (V.K.); kunic@veinst.hr (V.K.); prislin@veinst.hr (M.P.Š.); 2Laboratory for Diagnostics, Croatian Veterinary Institute, 32100 Vinkovci, Croatia; lolic@veinst.hr; 3Laboratory for Diagnostics, Croatian Veterinary Institute, 48260 Križevci, Croatia; sukalic@veinst.hr

**Keywords:** *Rotavirus A*, bovine coronavirus, cattle, gastroenteritis, interspecies transmission

## Abstract

Acute gastroenteritis (AGE) in cattle is a major economic concern, causing illness and reduced productivity. Bovine *Rotavirus A* (BoRVA) and bovine coronavirus (BCoV) are two key viruses responsible, leading to diarrhea and other health issues in cattle. While BoRVA mainly affects young calves, BCoV can impact cattle of all ages, causing digestive and respiratory clinical signs. This concise review examines the structure, spread, clinical signs, diagnosis, treatment, and prevention of BoRVA and BCoV, highlighting their potential to spread across species and the importance of effective management practices to control their impact.

## 1. Introduction

In recent years, the livestock industry worldwide has faced significant challenges from viral pathogens causing acute gastroenteritis (AGE) in cattle, with rotaviruses (RVs) and coronaviruses (CoVs) being major contributors to morbidity and economic losses. The prevalence of these viruses varies by species, region, and management practices but remains widespread in animal populations globally. For instance, the prevalence of bovine *Rotavirus A* (BoRVA) can reach up to 50% in young diarrheic calves in some areas, while bovine coronavirus (BCoV) prevalence can reach 70% in certain regions [1,2]. Although previous studies on the regional prevalence of BoRVA and BCoV have not specifically examined the relationship between cattle production systems and infection rates, it is likely that much of this research was conducted on intensive farms, as intensive farming is the dominant cattle production method globally [3]. Furthermore, intensive production systems are expected to have higher prevalence rates due to factors such as crowding, the constant introduction of new animals, and production-related stress, all of which increase the likelihood of viral transmission [4].

BoRVA and BCoV infections are linked to over 50% of pre-weaning calf mortality, with economic losses in countries such as Norway estimated at USD 10 million annually [5]. Besides higher mortality, the economic impact of these infections includes reduced weight or milk gain and increased veterinary costs. Feedlot calves shedding BCoV experience an 8.17 kg reduction in weight gain, and in adult cattle, BCoV-induced winter dysentery causes milk production drops of up to 70%, with an average loss of 51 L per cow per outbreak [6]. According to the Animal and Plant Health Inspection Service, the average cost of treating a single case of bovine respiratory disease in 2013 was USD 23.60, while digestive problems cost USD 9.90 per case [7].

AGE is a complex, multifactorial disease also influenced by environmental conditions, management practices, nutrition, and factors such as low temperatures, poor hygiene, colostrum deficiency, and individual animal susceptibility [5]. BoRVA and BCoV are common viral causes of bovine diarrhea and frequently co-occur. They may also be detected alongside other enteric pathogens like enteric viruses (bovine viral diarrhea virus, bovine torovirus, and bovine norovirus), protozoa (*Cryptosporidium*, *Coccidium* spp.), and bacteria, in particular *Escherichia coli, Clostridium perfrigens*, and *Salmonella* spp. However, this review will focus specifically on the main characteristics of BoRVA and BCoV infection, without addressing the complexities introduced by co-infections.

Additionally, BoRVA and BCoV are highly contagious and their infections cause the rapid onset of clinical signs, usually leading to malabsorption and maldigestion, often accompanied by relatively high mortality rates [2,4]

Although RVs and CoVs are primarily species-specific, sporadic cases of cross-species transmission have been observed. While the zoonotic potential of CoVs remains low, RVs are known to occasionally cross species and cause clinical signs in humans. Furthermore, *Rotavirus A* (RVA) in particular poses a risk, as it is responsible for over 128,500 deaths annually in young children [2,4,8]. Given the significant impact of BoRVA and BCoV on the cattle industry, additional research is crucial for a more comprehensive epidemiological insight and the efficient implementation of preventive strategies.

## 2. History, Taxonomy, and Structure of Rotavirus and Coronavirus

### 2.1. History, Classification, and Structure of Rotaviruses

RV was first identified in the 1970s and was initially classified as Nebraska calf diarrhea virus (NCDV). RV was officially named in 1974, referring to its round, wheel-like shape (Latin *rota* = wheel) when observed under an electron microscope (Figure 1a) [9]. RVs belong to the *Reoviridae* family and have a distinctive 100 nm diameter virion with an icosahedral capsid consisting of three protein layers but lacking an envelope. These double-stranded RNA viruses consist of eleven RNA segments ranging from 16 to 21 kb in size [10,11].

RVs are divided into nine species (RVA-RVD, RVF-RVJ) based on the antigenic similarities of the intermediate capsid protein VP6. RVA, RVB, RVC, and RVH species infect humans and other mammals, while RVD, RVF, and RVG infect avian species. RVI has been found in cats, dogs, and sea lions, and RVJ has been found in bats. An RVE species was once identified in domestic pigs but was removed from the official ICTV species list due to a lack of subsequent virus isolates and sequence data [11]. Most bovine RVs belong to the species *Rotavirus A*, and fewer in species *Rotavirus B* and *Rotavirus C* [2]. This distribution is reflected in a study conducted in Germany, where 59.1% of bovine fecal samples tested positive for RVA, 3% for RVB, and 6% for RVC, highlighting RVA as the most prevalent RV species in bovines [12].

RVs contain structural (VP1-4, VP6, VP7) and non-structural proteins (NSP1-6) (Figure 2a). Structural proteins are essential for virus particle formation, host specificity, and viral entry, while non-structural proteins support genome replication and interact with host proteins to induce immune responses [13]. Ten of the eleven RNA genome segments each encode one structural or non-structural protein, and the eleventh segment encodes two non-structural proteins, NSP5 and NSP6 [11].

RVA species can be classified using classical binomial nomenclature which recognizes G-genotypes (based on VP7 genome segment) and P-genotypes (based on VP4 genome segment). Up until now, 42 G and 58 P genotypes have been identified across all susceptible species [11,14]. However, whole-genome classification has become the standard, with specific nomenclature and cutoff values established for each genome segment [15]. This allows insights into strains’ similarities, origins, and reassortment events. This approach identified three human RVA genogroups: Wa-like, DS-1-like, and AU-1-like. The human AU-1-like genogroup likely originates from felines, while the Wa-like and DS-1-like genogroups may share origins with porcine and bovine RVs, respectively [16].

RV genetic diversity is driven by mechanisms such as single-point mutations, rearrangements, recombination, and reassortment events [17]. The latter, enabled by the segmented dsRNA genome, is a crucial driver of RV diversification and may impact vaccine efficacy, mainly when protection is based on specific G- and P-genotype responses [2]. Understanding viral structure is critical in the prevention of vaccine failure due to antigenic differences between vaccines and field strains. Continuous research is needed in developing polyvalent vaccines containing multiple RV genotypes for livestock and the improvement of heterologous vaccine protection [18,19].

**Figure 1 vetsci-11-00671-f001:**
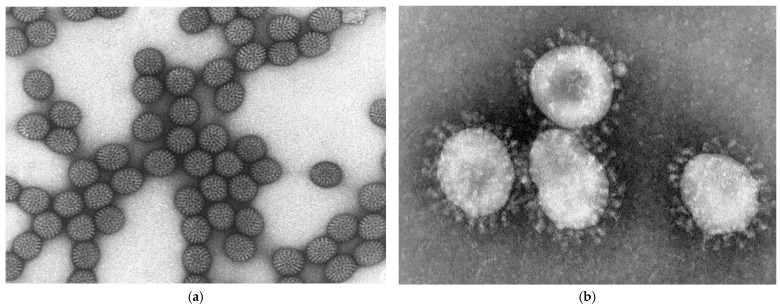
(**a**) Negative-staining electron microscopy reveals the characteristic wheel-like morphology of RV [20]. (**b**) A negative-staining electron micrograph of a CoV virion [21].

**Figure 2 vetsci-11-00671-f002:**
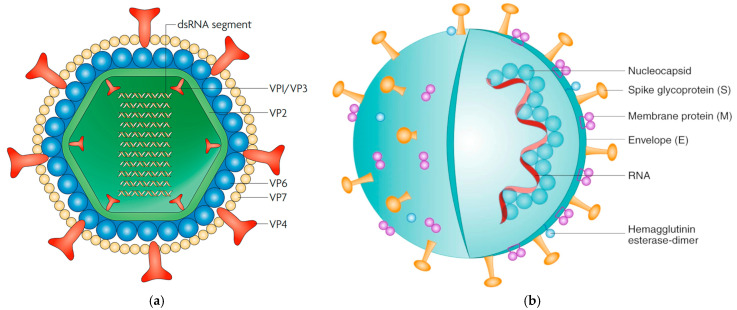
(**a**) A schematic of a triple-layered RV particle featuring the inner capsid (VP1, VP2, VP3), middle capsid (VP6), outer capsid (VP7), and spike protein (VP4) [22]. (**b**) A diagram of the CoV virion’s structure, featuring the nucleocapsid (N) protein and viral RNA genome at its core, encased in an envelope containing the spike (S), hemagglutinin–esterase (HE), membrane (M), and envelope (E) proteins [23].

### 2.2. History, Classification, and Structure of Coronaviruses

CoVs were first identified in the 1960s using electron microscopy, which revealed circumferential spikes on the virus surface, resembling a crown (Latin *corona* = crown) (Figure 1b) [24]. BCoV was later defined and confirmed as a common cause of calf diarrhea [25]. Taxonomically, CoVs are part of the *Coronaviridae* family, divided into four genera: *Alpha*-, *Beta*-, *Gamma*-, and *Deltacoronavirus*, each further divided into subgenera. Mammalian CoVs primarily belong to the *Alpha*- and *Betacoronavirus* genera. The *Alphacoronavirus* genus includes, for example, key porcine CoVs such as transmissible gastroenteritis virus (TGEV) and porcine epidemic diarrhea virus (PEDV). The *Betacoronavirus* genus includes severe acute respiratory syndrome CoV (SARS-CoV) in the *Sarbecovirus* subgenus and *Human CoV OC43* (HCoV-OC43) in the *Embecovirus* subgenus, both of which infect humans. In contrast, *Gamma*- and *Deltacoronaviruses* mainly infect avian species, with exceptions such as porcine CoV (HKU15) in the *Gammacoronavirus* genus and beluga whale CoV and bottlenose dolphin CoV (HKU22) in *Deltacoronavirus* genus. BCoV belongs to the *Betacoronavirus* genus, *Embecovirus* subgenus, and *Betacoronavirus gravedinis* species [26].

These enveloped viruses have a single-stranded, positive-sense RNA genome of about 30 kb, which is among the largest known viral RNA genomes [10]. BCoVs have a 65 to 210 nm diameter and a round shape with crown-like peplomers (Figure 1b) [4]. They contain five structural proteins—nucleocapsid (N), spike (S), membrane (M), envelope (E), and hemagglutinin–esterase (HE) proteins—along with 16 non-structural proteins (NSP 1-16) (Figure 2b) [4,10]. BCoV consists of a single serotype, with antigenic and genetic variations among strains [27]. BCoV is classified into two main genotypes: the classical genotype and the US wild ruminant genotype. These can be subdivided into 14 genotypes based on cutoff values similar to those used for RV classification [28]. Genotypes typically cluster based on geographic location and collection year; however, they do not determine whether BCoV leads to respiratory or gastrointestinal signs, implying that other factors may influence clinical outcomes [28].

CoVs exhibit high genetic diversity and host adaptability due to their large genomes, frequent homologous recombination during RNA replication, and mutations (mainly insertions and deletions) attributed to the low fidelity of RNA polymerase [4]. An example of this adaptability is the emergence of porcine respiratory CoV (PRCoV) from a deletion in the spike gene of transmissible gastroenteritis CoV (TGEV), which altered its virulence and tropism from the gastrointestinal to the respiratory system [10]. Additionally, CoVs are unique among RNA viruses in having developed a proofreading mechanism which enables them to avoid lethal error catastrophes during replication and contribute to quasispecies pools’ diversity [4].

## 3. Transmission and Epidemiology

### 3.1. Transmission Pathways and Epidemiological Dynamics of Rotavirus

RVs are highly prevalent, with nearly every domestic animal and bird species found to carry at least one native strain [10]. It is estimated that almost all children globally will have been infected by RV at least once by the age of five [29]. In cattle, BoRVA is a common cause of diarrhea in calves, particularly those between one and two weeks old. Young calves are highly susceptible to infection, as the buffering effect of milk helps BoRVAs survive a wide pH range. This protection allows the virus to endure the harsh conditions of the gastrointestinal tract and invade intestinal epithelial cells [30].

Viral shedding begins two days after clinical signs appear and can last for seven to eight days, continuously contaminating the environment [30]. While current knowledge of it in cattle is limited, studies in humans show that viral excretion can persist for up to two months, suggesting a similar possibility in cattle [31]. Although there are limited studies on viral shedding in asymptomatic adult cattle, similar conclusions could potentially be drawn from asymptomatic children infected with RVA, who exhibit a shorter duration of shedding and lower viral loads compared to those with RV diarrhea [32]. Transmission mainly occurs through the fecal–oral route, although saliva and respiratory transmission are also possible [33,34]. Cattle are infected by direct contact with symptomatic or asymptomatic individuals and contaminated objects, feed, or water [2]. RVs can retain their infectivity on various surfaces for up to 60 days, in food for 30 days, and in water generally for more than 60 days [35].

The mortality rate for RV diarrhea in newborn calves is generally between 5 and 20% but can increase under stressful conditions or due to secondary infections [30]. RVs are highly contagious, shedding up to 10^10^–10^12^ viral particles per milliliter of feces and can remain infectious for up to nine months at room temperature or one hour at 60 °C [2]. RVs are active year-round in tropical regions but peak in winter and spring in temperate climates [17].

The most common BoRVA genotypes are G6 (57%), G10 (21%), and G8 (4%). Among P-genotypes, P[5] (26%) is the most widespread, followed by P[11] (22%) and P[1] (2%). Nineteen G- and P-combinations have been identified in bovines, with G6P[5], G6P[11], and G10P[11] accounting for 40% of reports worldwide [19].

Despite mainly being species-specific, RVs can cross the species barrier and infect heterologous hosts. For example, a bovine-like strain with the G10P[11] genotype has been identified in humans in India, while bovine-like G8P[1] and G6P[1] strains have occasionally been found in children [17,36]. In addition to these possibly direct transmission events, there are also indications of potential reassortment events between RVA strains, including the detection of bovine-origin genotypes in humans, such as G8P[6], G8P[8], and G6P[6]. However, due to the low incidence of these possibly zoonotic genes and direct transmission events, it can be concluded that such occurrences are rare [17]. BoRVA genotypes have been identified in various animals as well, including G6P[14] in red deer and roe deer, G8P[14] in chamois, and G10P[15] in red deer, roe deer, camels, and sheep [37]. Possible reverse interspecies transmission has also been observed, with porcine genotypes (e.g., G3, G4, G5, G11, P[6], P[7]) and avian-origin strains (G17P[17], G18P[17]) being found in cattle [38,39]. However, collecting samples from wild animals presents significant challenges, making it difficult to assess the extent and significance of viral transmission between bovine RVs and wild animal populations. As a result, the dynamics and potential impact of such interspecies transmission remain largely unexplored.

### 3.2. Transmission Pathways and Epidemiological Dynamics of Coronavirus

Similarly to RVs, BCoV is globally spread, with cases confirmed in almost all continents. Seroprevalence studies indicate that over 90% of cattle are exposed to BCoV at some point [40]. While infection is usually age-unrelated, calves are most often affected within their first three weeks of life [10]. Interestingly, a recent study suggests that the age of intestinal mucus may influence the blocking of TGEV infection, a CoV found in pigs. The study may explain why younger animals are usually more susceptible to CoV infections [41]. While these findings primarily focus on pigs, it can be hypothesized that similar patterns may apply to BCoV infections in cattle.

BCoV is found in both the respiratory and intestinal tracts of symptomatic and asymptomatic cattle, with chronic subclinical infections in adult cattle serving as significant sources of the virus [4,27,42]. Like RVs, BCoV is mainly transmitted through the fecal–oral route; however, respiratory transmission is also possible [26]. Infection primarily spreads between calves or adult cattle, with environmental contamination posing an additional risk [6]. As an enveloped, single-stranded RNA virus, BCoV is less environmentally stable than RV but can remain infectious for up to three days in the presence of organic material. CoVs are also shown to bind effectively to porous materials such as wood, cement, clay, and charcoal [43].

BCoV is shed in nasal secretions and feces for up to five weeks post-infection, with some reports suggesting that shedding can persist for as long as 12 weeks. However, the likelihood of transmission during this extended period is minimal [27]. BCoV is more stable in lower temperatures and reduced sunlight, leading to a 50–60% increase in viral shedding during winter. Likewise, shedding rates rise by 65% during birth and 71% two weeks postpartum, increasing the risk of diarrhea in calves born from BCoV-positive cows [4]. Morbidity rates in affected herds can range from 20% to 100%, while mortality rates are 12% on average [10].

Due to its adaptability, high mutation rates, and recombination frequencies, BCoV has established distinct host-range lineages that have evolved separately over time. These include human CoV OC43 (HCoV-OC43), porcine hemagglutinating encephalomyelitis virus (PHEV), equine CoV (ECoV), murine hepatitis virus (MHV), rat sialodacryoadenitis virus, and canine respiratory CoV (CRCoV) [44,45]. Bovine-like CoVs have also been found in various domesticated and wild ruminants as well as dogs, raising the possibility that these animals could act as reservoirs for BCoV infections in cattle [46]. As with RVs, investigating and collecting samples for BCoV analysis from wild reservoirs is challenging, leaving the full extent and significance of viral transmission between wild and domestic animals unexplored. However, these documented cross-species infections can lead to the development of potentially recombinant strains that may evade immune responses and have the potential to spread to other species, including humans. For example, the human CoV HCoV-OC43 shows strong genetic and antigenic similarities to BCoV, suggesting that a zoonotic transmission event occurred relatively recently, around the late 19th century [4]. Progressive adaptive mutations in the hemagglutinin–esterase (HE) protein of HCoV-OC43 allowed the virus to lose its receptor-binding function and alter HE-mediated receptor destruction, resulting in the virus adapting to the human respiratory system [45].

## 4. Pathogenesis

### 4.1. Mechanisms of Infection and Pathogenic Effects of Rotavirus

As a leading cause of diarrhea in humans and animals, RVs were traditionally thought to target the gastrointestinal tract, infecting mature enterocytes and enteroendocrine cells in the small intestine [47]. However, emerging evidence suggests RV infection can also be systemic, extending beyond the intestinal lumen [33].

After entering a host organism through the previously mentioned routes, the virus attaches to targeted cells via its VP4 protein, which binds to cell surface receptors such as sialoglycans [13]. In less common cases, the virus may also bind to histo-blood group antigens (HBGAs), which are similar to the receptors found in humans [48]. After attachment, VP4 is cleaved into VP8 and VP5, allowing the virus to enter the cell through endocytosis or direct fusion with enterocytes [30,47]. Once inside, the virus begins replication and is eventually released from the cell through cell lysis or Golgi-independent non-classical vesicular transport [2]. After release, RVs can invade the intestinal lumen or enter the bloodstream and lymphatic system, circulating to various organs, including the liver, heart, lungs, kidneys, and central nervous system. RVs have been found in various immune cells including macrophages and B cells [49]. In a rat model, RVs were also detectable in blood serum for up to four days following the initial infection, confirming a short period of viremia [50]. However, blood cell-associated viremia appears to occur at low levels or in a small number of circulating cells [49]. Although RV’s presence outside the gastrointestinal tract is confirmed in both animals and humans, its full impact on other organs remains unclear [33].

Nonetheless, the most common clinical manifestation of RV infection in cattle is gastrointestinal distress, which results from several mechanisms. The destruction of enterocytes after viral replication reduces the absorptive surface area, causing unabsorbed glucose and the loss of electrolytes, which leads to osmotic imbalance and fluid accumulation in the lumen. Increased fluid secretion from intestinal crypts also leads to diarrhea and acidosis [51].

Another mechanism involves NSP4, which acts as a viral diarrhea-causing enterotoxin affecting nearby cells’ calcium channels [13]. Recent studies suggest an additional mechanism in which RVs exploit the host’s paracrine signaling by releasing adenosine diphosphate (ADP) from infected cells and activating P2Y1 receptors on adjacent uninfected cells. This activation triggers an intercellular calcium wave, which leads to fluid secretion. ADP also stimulates the release of serotonin and inflammatory mediators, further activating secretory reflex pathways in the gastrointestinal system [52].

### 4.2. Mechanisms of Infection and Pathogenic Effects of Coronavirus

On the other hand, BCoV infects epithelial cells in both the respiratory and gastrointestinal tracts, including nasal, tracheal and lung cells, as well as the villi and crypts of the intestines [27]. The virus uses its spike (S) protein and hemagglutinin–esterase (HE) protein for attachment and entry into host cells [40]. The S protein plays a critical role in determining tissue tropism; its S1 subunit binds to host cell receptors, such as sialic acid-containing receptors, inducing a conformational change that enables the S2 subunit to fuse with the host cell membrane [5,53]. Following fusion, BCoV can enter the cell directly or release its genomic RNA into the cytoplasm [40].

CoV’s RNA replication mechanism is unique compared to other positive-sense RNA viruses. Upon entering the cell, the part of viral RNA is translated into RNA-dependent RNA polymerase (RdRp), which synthesizes a complementary negative-sense RNA strand. This negative strand serves as a template for both the full-length viral genome and a set of 3′ co-terminal subgenomic mRNAs translated into various viral proteins. These proteins assemble at the endoplasmic reticulum and Golgi apparatus, accumulating in intracytoplasmic vesicles. The newly formed virions are then released via exocytosis, often leading to cell lysis or the formation of syncytia [10,40,54].

The pathological mechanisms of BCoV are similar to those of BoRVA, as both viruses cause enterocyte destruction, resulting in diarrhea and dehydration. However, while BoRVA is mainly limited to the small intestine, BCoV affects both the large and small intestines [51]. BCoV can also infect the respiratory tract, particularly targeting the nasal turbinates, trachea, and lungs. While CoVs are capable of systemic spread, BCoV generally remains localized in the respiratory or enteric system, although rare cases of systemic involvement have been reported [10]. For example, one study reported that a few calves with respiratory distress also showed neurological signs, with BCoV RNA detected in the brain, lungs, liver, and intestines [55]. The potential for the systemic spread of BCoV and its impact on different organ systems remains insufficiently investigated. The virus’s pathogenesis may start in the gastrointestinal tract and spread to other tissues through monocyte-associated viremia from the intestines or through cell-free viremia. Alternatively, it could begin in the respiratory system and spread to the intestines via swallowed mucus containing a high viral load [6,27,40,56]. While viremia in human CoVs can last up to 11 days, the duration of viremia for BCoV requires further research [57].

## 5. Clinical Signs and Pathoanatomical and Histopathological Lesions

BoRVA and BCoV-induced AGE present similar clinical signs, primarily causing diarrhea in young calves. However, BCoV can also cause winter dysentery in adult cattle and respiratory distress, either alongside diarrhea or as part of the bovine respiratory disease complex (BRDC) [4,10]. Both BoRVA and BCoV infections frequently occur alongside other pathogens, leading to co-infections that complicate the clinical presentation [58].

While CoVs are known for systemic spread in humans and animals, RVs have been traditionally seen as intestinal pathogens [10]. However, current knowledge shows that RVs can affect other organs like the central nervous system, liver, lungs, and other organs [33,59]. Despite this, BoRVA infections in cattle mainly present gastrointestinal distress [59].

Clinical signs in BoRVA infections usually appear after 12–24 h of incubation, ranging from mild to profuse watery diarrhea, which may be pale yellowish to greenish. Stool may contain mucus, undigested food, or, rarely, blood in cases of secondary bacterial infections. Additional signs include dehydration, weakness, fever, and emaciation [10,30]. In contrast, the incubation period in calves for BCoV-induced diarrhea ranges from one to seven days [6,40]. The severity of clinical signs depends on the level of maternal antibodies the calf ingests [40]. The incubation period may be extended in adult cattle, lasting up to eight days [6].

BCoV infection may present diarrhea similar to BoRVA infections (Figure 3a) but can also cause mild to moderate respiratory signs such as nasal and lacrymal discharge, coughing, dyspnea, and interstitial pneumonia [40,60]. Other clinical signs, such as neurological signs, are rarely observed [55]. Severe cases of BCoV infections may result in death within one to four days due to respiratory distress [60]. Diarrhea associated with both viral infections generally persists for about a week but can lead to fatal outcomes due to severe dehydration or shock [10].

While BoRVA and BCoV primarily target young calves, adult cattle can also be infected. BoRVA infections in adult cattle are typically subclinical; however, they can serve as reservoirs for transmission to calves [27]. BCoV in adult cattle may cause winter dysentery, characterized by bloody diarrhea (Figure 3b), anemia, and reduced milk production (up to 90%), but can also cause respiratory signs similar to those seen in calves [4,10].

Pathoanatomically, BoRVA-induced lesions in calves are typically not prominent but may include the thinning of the intestinal wall and liquid lumen contents (Figure 4a) [30,47]. BCoV-induced AGE presents similarly, including crypt epithelium necrosis and petechial hemorrhages (Figure 4b). In winter dysentery, blood clots or streaks may appear in the colon and rectum [56]. Respiratory involvement can lead to mucus and debris in the bronchi, interstitial pneumonia, and atelectasis [4,60,61]. Swollen, edematous mesenteric lymph nodes are common in both RV and CoV infections [58].

Histopathologically, both infections show villous atrophy, mononuclear cell infiltration in the lamina propria, and enterocyte vacuolization [13,30]. In BCoV infections with respiratory signs, tracheal hyperemia, epithelial erosion, and cellular infiltration in nasal and bronchial tissues are common [56,61].

## 6. Diagnosis

An accurate diagnosis is crucial for timely intervention in and the prevention of BoRVA- and BCoV-induced AGE. While diarrhea is the most prominent clinical sign of AGE, it is highly non-specific, making a diagnosis based solely on this clinical sign unreliable.

The optimal time for collecting fecal samples is during the clinical phase of the disease when the viral load is highest. It is recommended to collect fecal samples directly from animals, or in cases of diarrhea, to use rectal swabs for sample collection, while paying attention to cross-contamination between individual animals. In the case of an animal’s death, intestinal contents can be collected and sent for analysis. Stool containers for samples should preferably be sterile, and the usage of PPE is highly advised to avoid contamination during collection and transport. Samples should be stored in containers, cooled with ice packs (4 °C) during transport, and transferred to a laboratory to preserve pathogen viability and prevent nucleic acid degradation [5,62].

Real-time reverse-transcription polymerase chain reaction (real-time RT-PCR) and conventional RT-PCR are the preferred diagnostic methods due to their speed and sensitivity [5,33]. Given the high mutation rates of both viruses, continuous monitoring for sequence changes in target genes is essential to prevent primer incompatibility [5]. Fecal samples collected during the acute infection stage, prior to therapy, are commonly used for RT-PCR, though BCoV can also be detected in nasal secretions, even when gastrointestinal signs predominate [5,33]. Both RT-PCR assays typically target the VP2, VP6, and NSP3 genes for BoRVA and the M and N genes for BCoV [63,64,65,66]. In the field, LAMP PCR kits are also available to detect common enteric pathogens, including BoRVA and BCoV [67].

Rapid tests for field use, such as lateral flow immunochromatographic (IC) strip tests, are cost-effective tools for detecting pathogens in fecal samples; however, they have lower specificity and sensitivity compared to more advanced laboratory-based methods [62].

Other diagnostic methods, such as electron microscopy, tissue culture isolation, direct immunofluorescence, ELISA, immunohistochemistry (IHC), immunochromatography (IC), and latex agglutination, are less practical due to their time-consuming nature or, like AgELISA, because they offer lower specificity compared to RT-PCR [58,62,68]. Nevertheless, despite RT-PCR being the gold standard for BoRVA and BCoV diagnostics, these alternative methods are still utilized in various settings, such as for research purposes or situations requiring portability and cost-effectiveness [62]. Emerging third-generation nanopore sequencing holds promise as a cost-effective tool for detecting BoRVA and BCoV, as demonstrated in research on pigs [69].

Given the high likelihood of BoRVA and BCoV co-infections with other pathogens, differential diagnostics should include fecal bacterial culturing and fecal flotation along with direct microscopy to determine parasitic infections. Multiplex PCR protocols for detecting common enteric and respiratory pathogens are also practical in everyday use [5,70].

## 7. Treatment

Because there is no specific treatment for BoRVA and BCoV infections, the management of RV- and BCoV-induced AGE focuses on the management of clinical signs rather than targeting the virus directly. Initial treatment involves administering fluids and glucose to treat dehydration and correct acid-base and electrolyte imbalances caused by severe diarrhea [51].

To assess the level of dehydration, clinical signs such as an increased pulse and respiratory rate, decreased body temperature, dry mucous membranes, poor skin elasticity, the suckling reflex of the calf, and enophthalmos can provide us insights about the severity of the infection, while acidosis levels should be determined through laboratory methods [71,72].

Based on the severity of the infection, diarrheic cattle are categorized into two groups. Cattle with mild infections typically require oral electrolyte therapy, including sodium bicarbonate to correct acidosis, isotonic saline, a 2% glucose solution, and potassium. Severely dehydrated cattle need intravenous fluid therapy, which may include lactated or acetated Ringer’s solution, hypertonic saline, or isotonic/hypertonic sodium bicarbonate, as well as added potassium and 1–2% dextrose solution. The choice of fluids, their volume, and the administration rate depend on the age and weight of the animal, the severity and duration of clinical signs, and the level of metabolic acidosis [71,72].

In the case of winter dysentery, the condition typically resolves spontaneously within a few days without specific treatment. Nonspecific therapy includes providing fresh drinking water, palatable feed, and free-choice salt. However, in more severe cases, supportive therapies like oral or intravenous fluid therapy and blood transfusion may be necessary. NSAIDs and anti-hemorrhagic agents are used to control bloody diarrhea and prevent secondary bacterial infections [6,56].

Respiratory signs in BCoV infections should be treated with early parenteral antibiotics to prevent secondary pneumonia, along with NSAIDs such as meloxicam. Corticosteroids are generally not recommended as an adjunct therapy for undifferentiated pneumonia in feedlot cattle due to their potential to impair immune function. Other ancillary therapies such vitamin B or C injections, respiratory vaccines, antihistamines, anthelmintics, probiotics, and oral electrolytes may also be considered [56].

Co-infections with bacteria or protozoa in cattle may require antibiotics such as amoxicillin, ampicillin, or third- to fourth-generation cephalosporins. For severe dehydration, fluoroquinolones and IV fluid therapy are recommended. Protozoan infections like Cryptosporidiosis should be treated with halofuginone or azithromycin.

Along with these treatments, continued feeding with cow’s milk is advised for calves. With prompt supportive care, the prognosis of the infection is generally favorable [73].

## 8. Biosecurity Measures: Control and Prevention

Challenges during the diagnosis and treatment of viral AGE in cattle highlight the need for disease prevention strategies. Preventive measures include strict hygiene practices, herd management, and vaccinating pregnant dams with modified live or inactivated vaccines [10]. Hygiene and sanitation involve maintaining a clean environment, managing stress factors like low temperatures and drafts, and ensuring good ventilation [5].

The primary challenge in disinfecting farm surfaces is the presence of organic matter, which can interfere with the effectiveness of disinfectants. Therefore, thorough cleaning with detergent is essential before applying disinfectants. Common RV disinfectants include 6% sodium hypochlorite and phenol-based products, while CoVs, being enveloped viruses, can be effectively inactivated using ethanol (62–71%), hydrogen peroxide (0.5%), or sodium hypochlorite (0.1%) [43,74,75].

For a more practical approach, 1% Virkon S can be used on surfaces, equipment, and even in foot baths with a contact time of 3–10 min. Its broad-spectrum efficacy makes it effective against both enveloped and non-enveloped viruses, as well as a wide range of other pathogens commonly found in farm environments.

When designing cattle facilities, it is advisable to prioritize the use of non-porous materials such as metal, tiles, and vitrified cement, as these are easier to clean and disinfect [43]. In addition to using appropriate disinfectants, it is also recommended that veterinarians and individuals who interact with infected animals or contaminated environments consistently wear and regularly replace personal protective equipment (PPE).

Proper management includes adequate space for livestock, regular bedding removal, and grouping calves by age [5,10]. Maintaining dry and covered cattle housing areas is also critical, as the virus is more likely to remain viable in damp conditions [43]. Additionally, calving should be timed in favorable environmental conditions, and new herd members should be quarantined to prevent pathogen introduction [5].

Pregnant dams should receive proper care to enhance fetal development. Colostrum is crucial for calves, which should receive 3–4 L within the first six hours of birth [76]. Vaccinating pregnant dams during the last trimester boosts colostral antibody levels, providing passive immunity to calves [40]. The vaccination of young calves is generally avoided due to interference with maternal antibodies [40].

In addition to individual vaccines specifically targeting BoRVA and BCoV, there are also combined vaccines targeting common diarrhea pathogens authorized by the European Medicines Agency (EMA) [2,68,77]. Research also indicates that chicken egg yolk immunoglobulins (IgY) could protect calves from BoRVA and BCoV by delaying infection and reducing diarrhea severity, though further research is needed [78,79]. While usual individual vaccines for BCoV target enteric infections, intranasal vaccines authorized by the EMA may specifically address respiratory BCoV infections [77,80,81]. However, many BCoV vaccines do not protect adult cattle against winter dysentery and BRDC. A new multiepitope BCoV vaccine targeting spike (S) and nucleocapsid (N) proteins could offer broader protection across different age groups [42].

It is important to emphasize that the complete prevention of cattle’s exposure to BoRVA and BCoV is impossible due to their widespread nature. However, the risk of infection can be significantly reduced through the implementation of prophylactic measures [4,30].

## 9. Conclusions

BoRVA and BCoV are significant viral causes of AGE in cattle, mainly causing gastrointestinal distress in young calves, although all age groups can be affected. Despite differences in their RNA genomes, replication mechanisms, and molecular structures, both viruses represent significant threats to cattle health and productivity due to their ability to cause relatively high morbidity and mortality.

Their high mutation, reassortment, and recombination rates create the potential for interspecies transmission, which presents a possible risk to both animal and human populations. This risk is limited but more evident in BoRVA, while it remains only hypothetical for BCoV given its rare historical occurrences.

These factors, combined, highlight the need for effective preventive measures, such as strict hygiene, herd management, and vaccination strategies, to control the spread and impact of BoRVA and BCoV. This concise review may therefore be useful to veterinarians and other experts, offering practical and comparative insights into understanding and managing these infections.

## Figures and Tables

**Figure 3 vetsci-11-00671-f003:**
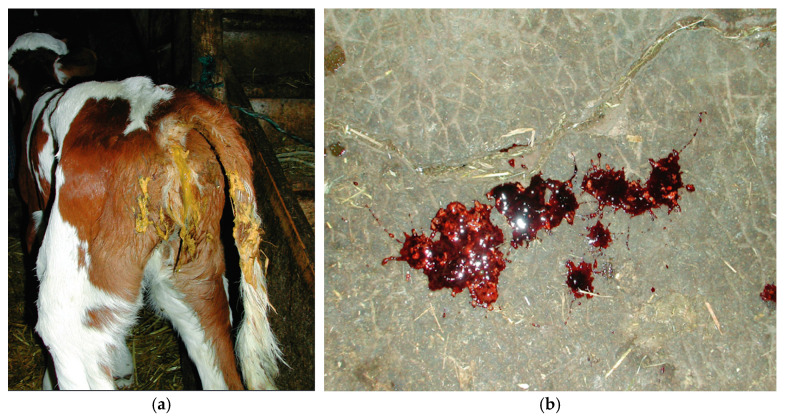
(**a**) A one-week-old calf exhibiting gastrointestinal signs, showing an example of yellowish diarrhea caused by BCoV [40]. (**b**) Hemorrhagic diarrhea resulting from BCoV infection [40]. Both images are courtesy of Cabinet Vétérinaire de Riom-es-Montagnes, Cantal, France.

**Figure 4 vetsci-11-00671-f004:**
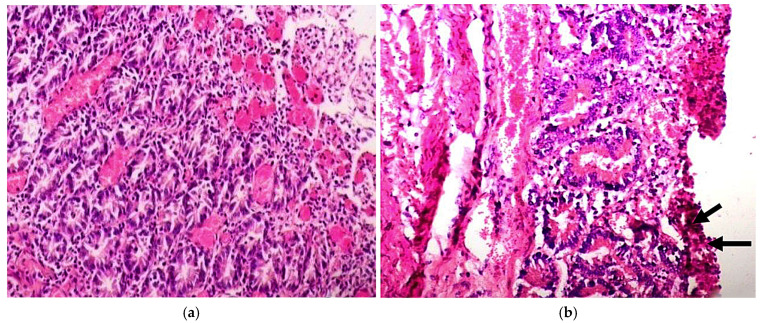
(**a**) A section of an intestine from a calf infected with RV, stained with H&E, reveals villous exfoliation, engorged capillary plexuses, and moderate infiltration of the lamina propria in the ileum [58]. (**b**) An intestinal section from a calf infected with BCoV presents thick homogeneous necrotic material covering the ulcerated mucosa of the colon, as indicated by the arrows [58].

## Data Availability

No new data created in this manuscript.

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
