# Peer review of "Comparative Insights into Acute Gastroenteritis in Cattle Caused by Bovine Rotavirus A and Bovine Coronavirus"

_vetsci, 2024, doi:10.3390/vetsci11120671_

Round 1
Reviewer 1 Report
Comments and Suggestions for Authors
Please take a look at the comments I've attached.

Author Response
Response to reviewers' comments
Reviewer 1
Comment 1: The manuscript is structured clearly and concisely, making it easy for veterinarians and cattle farmers to understand. However, it lacks how viral infections might interact with co-infections, such as bacteria and protozoa (lines 40-48), or the impact of environmental factors on disease dynamics—both of which play a crucial role in shaping disease outcomes. While diarrhea in cattle is briefly highlighted as a multifactorial issue involving bacterial and protozoal pathogens (lines 19 and 40), there’s little discussion on recognizing and managing co-infections. This lack is critical since co-infections can significantly influence disease presentation, complicate diagnosis, and ultimately affect treatment strategies. More attention to these aspects would provide a more comprehensive framework for addressing cattle infections with BoRVA and BCoV.
Response 1: We sincerely thank Reviewer 1 for recognizing the significance of our manuscript. In response to the comments provided, we have made the following additions: lines 64-66 in Section 1: Introduction, lines 343–345 in Section 5: Clinical Signs, Pathoanatomical and Histopathological Lesions; lines 434–438 in Section 6: Diagnosis; and lines 471–474 in Section 7: Treatment. Unfortunately, due to the concise nature of this review, which focuses only on BoRVA and BCoV infections in cattle, we are unable to explore in depth the complex relationships between various pathogens in cases of mixed infections. To address this limitation, we have acknowledged it in the manuscript and provided additional relevant references to guide interested readers toward further information.
Comment 2: Some sections' content is rich in a side-by-side comparison of different rotavirus (for example, lines 59-72) or BcoV (for instance, lines 117-132) but lacks a side-by-side comparison of BoRVA versus BCoV. For example, how do the genetic characteristics or clinical implications of BoRVA versus BCoV in causing acute gastroenteritis? How about the age- specific susceptibility of BCoV and BoRVA? For instance, BoRVA primarily affects calves under 14 days with gastrointestinal symptoms, while BCoV affects all ages, causing both gastrointestinal and respiratory distress. How about a direct comparison of transmission mechanisms (e.g., fecal-oral vs. respiratory routes) or epidemiological factors (e.g., seasonal variation, shedding dynamics) that would highlight the unique challenges each BCoV and BoRVA virus pose? This information could more strongly align with the study’s objective of offering actionable guidance to improve cattle management and mitigate the impact of acute gastroenteritis caused by these viruses.
Response 2: We thank the reviewer for their comment regarding the side-by-side comparison of BoRVA and BCoV. We would like to clarify that the manuscript contains this information in the relevant sections, however, given the constraints of this review due to conciseness, we have structured these comparisons within the context of the respective sections to maintain readability and focus. Specifically:
Genetic characteristics and clinical implications: Regarding BoRVA, we have highlighted the importance of structural variations in the species, particularly for epidemiological surveillance and vaccine development. Common bovine genotypes are mentioned in lines 199–202. However, differences in clinical presentations resulting from infections with various bovine genotype combinations have not been specifically addressed. This aligns with findings in human rotavirus infections, where different species and genotypes of RVA often produce similar clinical symptoms, despite genomic variations. To maintain the focus, we have decided not to explore these clinical distinctions further. Instead, we provide a general overview, recognizing that genotype diversity may be more relevant to vaccine design and epidemiology than to clinical management. Furthermore, in lines 155–160, we explain that clinical presentations cannot be determined based on the two genotypes of BCoV, reinforcing the notion that host factors, rather than viral genotype, primarily influence clinical outcomes.
Age-specific susceptibility: The differences in age-specific susceptibility are addressed in lines 175-176 and 224-225.
Transmission mechanisms and epidemiological factors: The manuscript compares the transmission routes (lines 188-189; 232-235) and epidemiological aspects, such as seasonal shedding dynamics, in lines 180–187 and lines 240–246.
Comment 3: While understanding the taxonomy and structure of BCoV and BoRVA has its merit, how this information is presented feels too complex to have much day-to-day practical value for cattle farmers or veterinarians (lines 57 to 132). The real question is, how does this knowledge help diagnose, prevent, or treat disease in cattle? For instance, pointing out how specific structural details, like the BCoV and BoRVA surface proteins, can improve diagnostic tools for identifying infections would make this information far more practical. Similarly, understanding the antigenic diversity of different BCoV (lines 117-118) and BoRVA strains (lines 66-67) could clarify how vaccines are developed and tailored to protect cattle against these viruses. Plus, knowing how these structural and genetic features explain why BCoV and BoRVA trigger gut issues in cattle would make it easier to connect science to the reality of managing transmission. Ultimately, any technical explanation is only helpful if it assists cattle farmers or veterinarians in solving problems and making better decisions for the cattle’s health.
Response 3: We thank the reviewer for their feedback on enhancing the discussion of taxonomy and structure to make it more practical and relevant for cattle farmers and veterinarians. We agree on the importance of connecting scientific insights to real-world applications and ensuring the presented information is both actionable and relevant. To address this, we have added a statement in lines 117–120 where the need for deeper knowledge about the structure of BoRVA (and by extension, BCoV) is stated. Furthermore, we have linked the described structural and non-structural proteins in Section 2: History, Taxonomy, and Structure of Rotavirus and Coronavirus with practical applications in Section 6: Diagnosis. Here, various proteins are highlighted as target genes for PCR methods (lines 417–418). Additionally, VP4 and VP7 structural proteins of RVA are discussed as tools for genotyping (lines 103–112) and are underlined as crucial in the context of vaccine development, as reflected in the new lines we added to your suggestion (lines 117–120).
Comment 4: While the section provides an in-depth understanding of transmission and epidemiology (section in line 133), it needs more straightforward, actionable suggestions for veterinarians and cattle farmers. For instance, the authors highlighted that the transmission of both BoRVA and BCoV mainly occurs via the fecal-oral route (lines 148 and 184). How do vaccination strategies, biosecurity measures, or hygiene protocols help in breaking the fecal oral transmission cycle? Include practical prevention and control measures, such as disinfection protocols, vaccination, or management practices to reduce environmental contamination and viral spread. This type of discussion would make the information more practical.
Response 4: We thank the reviewer for their suggestion, we agree that including more practical prevention and control measures strengthens this section. To address this, we have expanded Section 8: Biosecurity Measures; Control and Prevention by incorporating detailed recommendations. These include the use of proper disinfectants (lines 484–493) and practical design considerations to optimize disinfectant efficiency (lines 494–497). Additional management strategies are outlined in lines 497–504. Furthermore, vaccination strategies are discussed in lines 507–509 and 510–519.
Comment 5: Although interspecies transmission and systemic spread were mentioned (lines 133-211), the conclusion should specifically call attention to areas that require further research, such as BCoV’s (lines 190-196) or RV (lines 151-156) impact on winter inflammation of the intestines and especially the colon, which leads to severe diarrhea containing blood and/or mucus or improved diagnostics for both pathogens. Moreover, while cross-species infections and zoonotic potential are outlined, the impact of these findings on cattle management is not fully explored. The authors should highlight the risks of back-transmission from wild reservoirs or domestic animals, which would make the interspecies transmission and systemic spread more relevant to cattle producers.
Response 5: We thank the reviewer for their suggestion on emphasizing areas for further research. We fully agree and, while preparing this review, identified several key knowledge gaps. These include improving heterologous vaccine protection (118-120), developing vaccines for adult cattle (517-519), investigating interspecies transmission with a focus on wild reservoirs (215-219; 257-259), exploring the systemic spread and viremia duration of these viruses, and further exploration of the pathogenesis mechanisms of both pathogens. Additionally, we highlight the need to explore the reasons behind varying clinical manifestations of BCoV and the zoonotic potential of both BoRVA and BCoV. These areas require further data to advance our understanding of these pathogens, and we believe the valuability of this review also lies in its possible inspiration to researchers to pursue these critical areas.
Comment 6: While the pathogenesis section is rich of information (line 212), the section fails to include a comparative discussion of BoRVA and BCoV pathogeneses, which is essential given the study’s objective is to provide a comparative analysis of the two viruses. RVs were traditionally target the gastrointestinal tract, and can also be systemic (lines 214-217). On the other hand, BCoV infects both the respiratory and gastrointestinal tracts (lines 242-245). Adding such a discussion would help readers better understand how the two viruses differ or overlap in their pathogenic effects.
Response 6: We thank the reviewer for this comment. As stated in the response 2, the manuscript contains this information in the relevant sections, speciffically in lines 322-326. However, given the concise nature of this review, we have structured these comparisons within the context of the respective sections to maintain readability and focus.
Comment 7: The diagnosis section must be discussed thoroughly to align with this manuscript’s practical objectives (lines 320-338). Here is a list of suggestions the authors should address: (1) While using fecal samples and nasal secretions for many cases is mentioned (lines 327-330), there is little detail on proper sample collection, storage, and handling. Including this guidance would make the advice more practical for veterinarians. Explaining the need for sterile containers, refrigeration, or transport timelines could help preserve sample integrity. (2) While the section compares RT-PCR to other techniques like electron microscopy and AgELISA (lines 324-338), it does not explore the limitations of these assays in depth or explain when they may still be relevant (e.g., for research purposes or low-cost diagnostics). (3) Diarrhea in cattle is linked to multifactorial causes, including bacterial and protozoal pathogens (lines 19 and 40). More mention needs to be made on how to rule out or address co-infections, which can alter disease presentation and complicate diagnosis.
Response 7: We thank the reviewer for their detailed feedback and valuable suggestions for improving the section 6: Diagnosis. We have revised the section to include the requested data in lines 402-410 regarding the sample collection and transportation, the use of additional diagnostic methods in lines 421-431, and addressed co-infections in lines 434-438.
Comment 8: The authors highlight that the treatment for RV- and BCoV-induced AGE focuses on symptom management rather than targeting the virus directly, and treatment options are listed (lines 341-349). There is limited detail on dosage, administration routes, or treatment duration. Providing specific examples or guidelines for fluid replacement therapy or antibiotic use would make this section more actionable. The section should differentiate between mild, moderate, or severe cases, which could help veterinarians prioritize treatment strategies. For example, severe dehydration may require intravenous fluids, whereas mild cases may benefit from oral rehydration alone. The range of suggested interventions, such as antibiotics, vaccines, antihistamines, and vitamins, is broad but lacks a structured explanation or prioritization. For example, the authors indicated that respiratory symptoms in BCoV infections should be treated with early parenteral antibiotics to prevent secondary pneumonia, along with NSAIDs, vitamin B or C injections, respiratory vaccines, antihistamines, anthelmintics, probiotics, and oral electrolytes (lines 346-348). This could lead to confusion about which treatments are essential versus supplemental. Prophylactic measures (e.g., vaccination or management practices to reduce disease spread) are not included despite their importance in reducing infection rates and severity.
Response 8: We thank the reviewer for their feedback. We have revised Section 7: Treatment to include the requested details, specifically the indications for fluid therapy (lines 446–450), methods of fluid therapy administration (lines 451–458), and specific antibiotics. However, dosage and administration routes were not discussed, as these vary by manufacturer. The section has been restructured to address essential and supplementary treatments (lines 459–470), and specific treatments for common co-infections have been added (lines 471–474). Prophylactic measures are further discussed in Section 8: Biosecurity Measures; Control and Prevention.
Comment 9: While the conclusion is concise (lines 378-391), it could briefly highlight some of the article's most practical or novel insights, such as differences in clinical presentation, diagnostic approaches, or vaccine developments. The call to implement preventive measures could be more specific. For example, highlighting high-priority actions, such as vaccinating dams in the last trimester for BoRVA and improving winter-specific management for BCoV, would provide more focused guidance.
Response 9: We thank the reviewer for their suggestion; however, we would prefer to keep the conclusion concise, given the nature of this manuscript. Instead, we direct readers to specific sections where the most critical data have been thoroughly discussed.
We appreciate the reviewer’s contribution, as their additions and suggestions enhanced this concise review and provide clear, actionable recommendations for veterinarians and cattle producers.

Reviewer 2 Report
Comments and Suggestions for Authors
It is a good job done about these viruses but there is not given the importance of the viruses or new knowledge to make it interesting for the readers.
you should give more importance to the recent articles published instead of giving only the references in the text. I mean, you could start with according with this publication in 2024....
I will give more importance to the fact that it could be a zoonosis and you can give more data about the economic losts.
Author Response
Response to reviewers' comments
Reviewer 2
Comment 1: It is a good job done about these viruses but there is not given the importance of the viruses or new knowledge to make it interesting for the readers.
you should give more importance to the recent articles published instead of giving only the references in the text. I mean, you could start with according with this publication in 2024....
I will give more importance to the fact that it could be a zoonosis and you can give more data about the economic losts.
Response 1: We thank the reviewer for their valuable feedback and suggestions which have contributed to improving the manuscript. We appreciate the recognition of the work done on these viruses and agree that emphasizing recent research and economic impacts could increase the manuscript's relevance for readers.
The purpose of this review was to compile and present the current knowledge in a practical and concise manner, while offering a comparative analysis of these two enteric pathogens. All relevant sections were addressed in a comparable way to highlight the similarities and differences between the two viruses. Due to the frequent occurrence of co-infections, reviews focusing on both, BoRVA and BCoV are less common, making this article, to the best of our knowledge, unique and useful to both professionals and the general public.
In the process of writing this review, we have also identified and highlighted several knowledge gaps and areas that require further investigation, such as improving heterologous protection of vaccines, developing vaccines for adult cattle, exploring interspecies transmission with a focus on wild reservoirs, exploring the systemic spread and viremia duration of these viruses, and further studying the pathogenesis mechanisms of both pathogens. We also emphasize the need for research into the underlying reasons behind the varying clinical manifestations of BCoV, as well as further exploration of the zoonotic potential of BoRVA and BCoV. These areas highlight the need for continued research and further data to advance our understanding of these pathogens, and this review may inspire researchers to pursue these areas of study.
Regarding the reviewer’s suggestions; we acknowledge that the economic impact of these viruses was not sufficiently emphasized and have therefore added relevant data (lines 48-56) on the economic costs associated with BoRVA and BCoV infections, including their impact on livestock productivity and veterinary care. This addition will demonstrate to readers, in practical terms, why biosafety and preventive measures are important when dealing with these pathogens.
We have carefully incorporated references to more recently published studies to ensure that the manuscript reflects the latest developments in the field.
While we understand the importance of zoonotic potential of these pathogens, we have intentionally chosen not to emphasize this aspect in detail due to the rare occurrence of zoonotic transmission of these viruses. However, we did acknowledge the possibility of zoonotic transmission (lines 71-72; 203-210; 259-267) and have mentioned it emphasizing its relatively low potential.
Once again, we thank the reviewer for their constructive comments and hope that we have addressed their concerns while maintaining the focus of the manuscript on its primary objectives.
Reviewer 3 Report
Comments and Suggestions for Authors
Throughout the document, the word symptoms should be changed to signs or clinical signs
Lane 37-39, mention in the text how this distribution is presented depending on the characteristics of the production systems, with the intention of presenting information that associates animal density and the prevalence of both viruses.
Lane 71-72, mention the cases where RVB and RVC have been reported.
Lane 102-110, it is recommended that in each genus you mention one or two examples of the most representative viruses and the species they affect.
Lane 144, indicate a time interval in which viral excretion lasts.
Lane 149-150, indicate the time that RV maintains its infectivity in objects, food or water.
Lane 219, mention the cellular receptor involved
Lane 223-226, indicate the duration of viremia and whether the viral presence is associated with cells or free in blood.
Lanes 228-232, specify the age at which the described post-infection alterations occur.
Lanes 246, mention the cellular receptor that the virus uses to infect susceptible cells
Lanes 259-271, indicate the time of excretion and viremia
Lanes 315-318, it would be advisable to present images of the most representative microscopic lesions
Lanes 321-338, mention IHC and serologic assays.
Lanes 340-349, this entire section describes palliative and, in many cases, non-specific treatments. The section should begin with the statement: There is no specific treatment for RV- and BCoV infections, then describe the support alternatives that can be used; however, mention the prognosis when implementing them.
Lanes 352-360, mention effective disinfectants for both viruses or management using other procedures that favor disinfection, in addition to stricter hygiene habits.
It should be noted that the use of personal protective equipment for workers or people who have contact with infected livestock is useful to reduce the potential zoonotic risk of these two infections.
Author Response
Response to reviewers' comments
Reviewer 3
Comment 1: The word "symptoms" should be changed to "signs" or "clinical signs"
Response 1: We appreciate this suggestion and have revised the manuscript to replace "symptoms" with "signs" and "clinical signs" throughout the text to ensure clarity and consistency.
Comment 2: Lane 37-39, mention how this distribution is presented depending on the characteristics of the production systems, associating animal density and the prevalence of both viruses
Response 2: We thank the reviewer for this valuable suggestion. We have addresed the lack of detailed information about the distribution of BoRVA and BCoV infections across different production systems in lines 40–47.
Comment 3: Lane 71-72, mention the cases where RVB and RVC have been reported
Response 3: We thank the reviewer for this comment; we have included more information about the lower prevalence of RVB and RVC in cattle in lines 93-96.
Comment 4: Lane 102-110, mention one or two examples of the most representative viruses in each genus and the species they affect
Response 4: We appreciate this suggestion, we have addresed it in the lines 138-146, where we provided examples of representative viruses from the genera within the Coronaviridae family.
Comment 5: Lane 144, indicate a time interval in which viral excretion lasts
Response 5: Upon further research, we have decided to state that viral excretion lasts 7–8 days, with the possibility of being prolonged (lines 181-183). This conclusion is based on human research, as limited data is available regarding cattle.
Comment 6: Lane 149-150, indicate the time that RV maintains its infectivity in objects, food, or water
Response 6: We thank the reviewer for this comment, we have added the requested data in lines 191-192 and included additional information on BCoV infectivity in lines 236-239 in Section 3.2 to maintain the comparative nature of this manuscript.
Comment 7: Lane 219, mention the cellular receptor involved
Response 7: We have included a reference to the specific cellular receptors involved in the entry of BoRVA into host cells in the lines 275-278.
Comment 8: Lane 223-226, indicate the duration of viremia and whether the viral presence is associated with cells or free in blood
Response 8: We have revised this section to provide information on the duration of viremia in cattle infected with BoRVA and to specify whether the virus circulates in the blood as free particles or in association with infected cells. Due to the limited data available on cattle, relevant conclusions were drawn from a study using a rat/mouse model, as outlined in lines 284-288.
Comment 9: Lanes 228-232, specify the age at which the described post-infection alterations occur
Response 9: As stated in line 351, the first clinical signs appear after 12–24 hours of RV incubation. Unfortunately, there is no specific data pinpointing the exact timing of microscopic and macroscopic alterations. However, it can be concluded that the initial signs emerge within this period, with more severe symptoms developing over time.
Comment 10: Lane 246, mention the cellular receptor that the virus uses to infect susceptible cells
Response 10: We have included a reference to the specific cellular receptors involved in the entry of BCoV into host cells in the lines 309-312.
Comment 11: Lanes 259-271, indicate the time of excretion and viremia
Response 11: While the duration of BCoV excretion is specified in lines 240–241, lines 332–337 have been updated to include information on the duration of viremia for BCoV infection. However, due to the limited research on BCoV viremia in cattle, a parallel was drawn with human CoVs.
Comment 12: Lanes 315-318, it would be advisable to present images of the most representative microscopic lesions
Response 12: We thank the reviewer for the suggestion, we have added figures 4a and 4b in lines 391-395.
Comment 13: Lanes 321-338, mention IHC and serologic assays
Response 13: We appreciate this comment, we have revised this section to include IHC and serologic assays as diagnostic tools for detecting BoRVA and BCoV infections in lines 425-428. We have also added the information about the rapid tests in lines 421-424, which is particularly useful for field applications.
Comment 14: Lanes 340-349, start the section with: "There is no specific treatment for RV- and BCoV infections," then describe the support alternatives and mention prognosis
Response 14: We thank the reviewer for the suggestion, we have added a similar statement at the beginning of the section in lines 441–443. Furthermore, we have expanded the details regarding supportive therapy, including fluid therapy, in lines 451–458. Additionally, we have provided more detailed information on the treatment of winter dysentery and respiratory signs in lines 459-464 and 465-4670 Finally, a statement regarding the prognosis when supportive care is applied has been included in line 476.
Comment 15: Lanes 352-360, mention effective disinfectants for both viruses or management procedures that favor disinfection, in addition to stricter hygiene habits
Response 15: We have slightly rewritten this section to include new information, including the usage of proper disinfectants and a practical suggestion for disinfection (lines 484-493).
Comment 16: The use of personal protective equipment for workers or people who have contact with infected livestock is useful to reduce potential zoonotic risk
Response 16: We have added a recommendation for veterinarians and workers who interact with animals regarding the use of PPE in lines 497–499. Additionally, in lines 520–522, we included a statement highlighting that while BoRVA and BCoV cannot be completely eradicated, the risk of infection can be minimized with proper prophylactic measures.
We would like to thank the reviewer once again for their helpful suggestions and hope that this revised manuscript meets their expectations.
Round 2
Reviewer 2 Report
Comments and Suggestions for Authors
You have given a good change and increase of the zoonotic part and new references. Now it is more atractive and useful for readers.
Once again, good job.